# Mechanical Loading-Driven Tumor Suppression Is Mediated by Lrp5-Dependent and Independent Mechanisms

**DOI:** 10.3390/cancers13020267

**Published:** 2021-01-13

**Authors:** Yan Feng, Shengzhi Liu, Rongrong Zha, Xun Sun, Kexin Li, Alexander Robling, Baiyan Li, Hiroki Yokota

**Affiliations:** 1Department of Pharmacology, College of Pharmacy, Harbin Medical University, Harbin 150081, China; fengya@iu.edu (Y.F.); zhar@iu.edu (R.Z.); sunxun@iu.edu (X.S.); kexli@iu.edu (K.L.); 2Department of Biomedical Engineering, Indiana University Purdue University Indianapolis, Indianapolis, IN 46202, USA; liu441@iu.edu; 3Department of Anatomy Cell Biology and Physiology, Indiana University School of Medicine, Indianapolis, IN 46202, USA; arobling@iupui.edu; 4Indiana Center for Musculoskeletal Health, Indiana University School of Medicine, Indianapolis, IN 46202, USA; 5Simon Cancer Center, Indiana University School of Medicine, Indianapolis, IN 46202, USA

**Keywords:** breast cancer, prostate cancer, mechanical stimulation, osteocyte, Lrp5, chemerin, nexin

## Abstract

**Simple Summary:**

Advanced breast cancer and prostate cancer metastasize to varying organs including the bone. We show here that mechanical loading to the knee suppresses tumor growth in the loaded bone and the non-loaded mammary pad. Although lipoprotein receptor-related protein 5 (Lrp5) in osteocytes is necessary to induce loading-driven bone formation, loading-driven tumor suppression is regulated by Lrp5-dependent and independent mechanisms. Lrp5 overexpression in osteocytes enhances tumor suppression, but without Lrp5 in osteocytes, mechanical loading elevates dopamine, chemerin, p53, and TNF-related apoptosis-inducing ligand (TRAIL) and reduces cholesterol and nexin. Their systemic changes contribute to inhibiting tumors without Lrp5. Osteoclast development is also inhibited by the load-driven regulation of chemerin and nexin.

**Abstract:**

Bone is mechanosensitive and lipoprotein receptor-related protein 5 (Lrp5)-mediated Wnt signaling promotes loading-driven bone formation. While mechanical loading can suppress tumor growth, the question is whether Lrp5 mediates loading-driven tumor suppression. Herein, we examined the effect of Lrp5 using osteocyte-specific Lrp5 conditional knockout mice. All mice presented noticeable loading-driven tumor suppression in the loaded tibia and non-loaded mammary pad. The degree of suppression was more significant in wild-type than knockout mice. In all male and female mice, knee loading reduced cholesterol and elevated dopamine. It reduced tumor-promoting nexin, which was elevated by cholesterol and reduced by dopamine. By contrast, it elevated p53, TNF-related apoptosis-inducing ligand (TRAIL), and chemerin, and they were regulated reversely by dopamine and cholesterol. Notably, Lrp5 overexpression in osteocytes enhanced tumor suppression, and osteoclast development was inhibited by chemerin. Collectively, this study identified Lrp5-dependent and independent mechanisms for tumor suppression. Lrp5 in osteocytes contributed to the loaded bone, while the Lrp5-independent regulation of dopamine- and cholesterol-induced systemic suppression.

## 1. Introduction

Breast cancer is the second leading cause of cancer death in women worldwide. While one out of eight women suffers from breast cancer in her lifetime, treating advanced breast cancer remains a challenge [1]. Prostate cancer is the most frequent tumor found in men worldwide [2]. Bone metastasis is a frequent complication [3] for bone metastatic breast cancer and advanced prostate cancer [4,5]. Metastasis to bone induces pain, hypercalcemia, and the risk of bone fracture [6]. It is well known that bone is responsive to its mechanical environment and that a lack of mechanical stimulation promotes bone loss [7,8]. Loading modalities such as axial loading and whole-body vibration can increase bone mass by activating bone-forming osteoblasts [9,10], as well as inhibiting bone-resorbing osteoclasts [6]. Mechanical loading also suppresses tumor-driven osteolysis by restraining bone resorption [11], inhibits the progression of myeloma in the bone [12], and induces apoptosis of breast cancer cells [13]. The molecular mechanisms of loading-driven tumor suppression and the role of Wnt signaling remains elusive.

We previously reported that osteocytic expression of low-density lipoprotein receptor-related protein 5 (Lrp5), a co-receptor for Wnt ligands, is necessary to induce loading-driven bone formation [14,15,16]. It is also reported that specific gain-of-function missense mutations in Lrp5 induced protection from disuse-related bone loss by increasing osteogenic responsiveness to loading [17]. Using a mouse model of bone metastasis associated with breast cancer and prostate cancer [18], the main question we addressed in this study was whether Lrp5 in osteocytes is necessary to induce loading-driven tumor suppression. We employed mice with osteocyte-selective deletion of Lrp5 and evaluated the Lrp5-dependent and independent mechanisms involved in loading-driven tumor suppression.

Cyclic lateral loads were applied to the knees of wild-type and osteocyte-specific Lrp5 knockout mice. It is reported that knee loading not only stimulates bone formation and prevents cartilage degeneration but also promotes vessel remodeling and bone healing in the necrotic femoral head [19]. Knee loading is shown to activate endocrine and neural signaling and induce local and global responses [20,21], and we have shown that it can inhibit tumor-driven osteolysis in the mouse model of breast cancer and prostate cancer [22]. However, little is known about the role of Lrp5-mediated Wnt signaling in loading-driven tumor suppression. In this study, we examined the potential involvement of Lrp5 in osteocytes in tumor suppression at a locally loaded site of the tibia as well as at a remote non-loaded site of the mammary fat pad.

In evaluating the Lrp5-mediated and Lrp5-non-mediated mechanisms of loading-driven tumor suppression, we examined the expression of oncogenic genes such as Runx2, Snail, and TGFβ in the context of Lrp5 overexpression and silencing. In our previous work, step aerobics was used as a means of mechanical stimulation, which significantly altered the levels of dopamine and cholesterol in human urine samples [23]. Thus, we examined their urinary and serum levels in wild-type and knockout mice before and after knee loading. We also determined the role of tumor-promoting cytokines and metalloproteinases such as nexin (Serpin peptidase inhibitor, clade E, member 2 -SERPINE2), MMP2, MMP3, and MMP9, as well as tumor-suppressing factors such as p53, TNF-related apoptosis-inducing ligand (TRAIL), and chemerin (retinoic acid receptor responder protein 2 (RARRES2)), and an apoptosis marker, cleaved caspase 3.

Beyond evaluating suppression of tumor progression, we also measured the effects of lateral loading on the development of bone-resorptive osteoclasts by determining the expression of NFATc1, a master transcription factor for osteoclastogenesis, and cathepsin K, a proteinase for osteoclast maturation. Tumor-induced osteolysis is amplified by the vicious cycle through tumor-osteoclast interactions [11]. The results herein supported that both Lrp5-dependent and independent mechanisms are present in loading-driven tumor suppression. Since Lrp5-mediated Wnt signaling can promote tumorigenesis and tumor progression, the tumor-suppressing role of Lrp5 in osteocytes indicates that osteocytes and tumor cells use Lrp5 signaling in distinct and opposing ways.

## 2. Results

### 2.1. Inhibitory Effects of Knee Loading on Mammary Tumors

The main aim of this study was to identify the role of Lrp5 in inducing loading-driven suppression of tumor progression. We first confirmed genotypes for the presence of Cre and Lrp5 floxed alleles. Mice lacking Lrp5 in osteocytes (KO mice) had Dmp1-Cre and homozygous flox alleles (Figure 1A). Using mice evenly distributed by age into four groups (Figure 1B), we evaluated the effect of daily knee loading for two weeks (1 N peak-to-peak at 2 Hz for 5 min) on mammary tumors that were induced by the inoculation of EO771 cells into the mammary fat pad of C57BL/six female mice (Figure 1C). Knee loading reduced the tumor size and weight in the mammary fat pad in mice regardless of Lrp5 deletion (Figure 1D; Appendix A). The result revealed that the mammary tumor can be suppressed remotely by mechanical stimulation to the knee both in wild-type and knockout mice independent of osteocytic Lrp5.

### 2.2. Protection of Tumor-Invaded Tibia by Knee Loading

We next examined the effect of knee loading on the tumor-invaded tibia. After intratibial injection of EO771 mammary tumor cells to female mice and transgenic adenocarcinoma of the mouse prostate (TRAMP) prostate tumor cells to male mice, we applied knee loading to both wild-type and knockout mice using the same loading condition in the previous experiment. We observed that daily knee loading reduced the degradation of trabecular bone in the proximal tibia (Figure 1E). Specifically, a significant increase in bone volume ratio (BV/TV), trabecular number (Tb.N), and bone mineral density (BMD) was observed by knee loading, with a reduction in the trabecular separation (Tb.Sp) (Figure 1F; Appendix A). The result of these four parameters is consistent with the protective effect of knee loading. Also, Hematoxylin and Eosin (H&E)-stained sagittal sections revealed that knee loading reduced the area of tumor invasion regardless of the deletion of Lrp5 in osteocytes (Figure 2A–C). Notably, although knee loading suppressed the growth of tumors in the tibia, the tumor-induced damage was more severe when Lrp5 was deleted. The loading effect on tumor areas was larger in wild-type mice than knockout mice in both females and males (Figure 2D). These results indicated that Lrp5 in osteocytes contributed to protecting bone from tumor invasion, but loading-driven suppression of tumor-induced bone loss was observed even in osteocyte-specific Lrp5 knockout mice.

The results so far showed the beneficial role of Lrp5 in osteocytes in the tumor-invaded tibia regardless of knee loading. In the absence of tumor inoculation, the benefit of knee loading to bone formation was only observed in wild-type mice. While knee loading elevated BV/TV, BMD, Tb.N with a decrease in Tb.Sp in the trabecular bone of wild-type mice, it did not induce any significant alteration in knockout mice (Figure 3A,B). Furthermore, in the cortical bone of the same proximal tibiae, knee loading increased BMD in the tibia of wild-type mice but not that of knockout mice (Appendix A). The result indicates that Lrp5 in osteocytes is necessary for inducing loading-driven bone formation, but it is not required for causing loading-driven tumor suppression.

### 2.3. Loading-Dependent Alterations in Dopamine and Cholesterol in the Serum and Urine

To address the mechanism of loading-driven tumor suppression in the absence of Lrp5 in osteocytes, serum and urine samples were collected from female mice before and 1 h after the application of knee loading and assayed for dopamine and cholesterol. Compared to the pre-loading samples, the post-loading samples exhibited elevated levels of dopamine and reduced the level of cholesterol in the serum and urine (Figure 3C–F). Notably, the altered levels of dopamine and cholesterol were observed irrespective of Lrp5 in osteocytes.

### 2.4. Regulation of Chemerin as a Tumor Suppressor and Nexin as a Tumor Promoter

Besides the loading-driven alterations in cholesterol and dopamine, antibody-based cytokine array analysis revealed that knee loading substantially elevated the relative expression level of chemerin and reduced the relative expression level of nexin in the serum of both wild-type and knockout mice (Figure 4A, Appendix A). The regulation of chemerin and nexin was also observed in EO771 cells in response to dopamine and cholesterol (Figure 4B, Appendix A). The summary of four pairs of cytokine profiles (knee loading in wild-type and knockout mice, and the responses to dopamine and cholesterol) indicated that chemerin acted as a tumor suppressor, while nexin was considered as a tumor promoter (Figure 4C). Of note, MMP2 and MMP3 also acted as tumor promoters in five out of eight cases (Figure 4C). Western blot analysis showed that the level of chemerin was elevated by dopamine and reduced by cholesterol in EO771 cells (Figure 4D). Furthermore, chemerin was upregulated and nexin was downregulated by knee loading in the serum of wild-type and knockout mice (Figure 4E). Chemerin reduced MTT-based cell viability, Transwell invasion, and scratch-based migration in EO771 cells, while nexin elevated them (Appendix A). Collectively, the result supported the possibility that knee loading suppressed tumor progression in part by systematically elevating dopamine and chemerin, while reducing cholesterol and nexin, even in the absence of Lrp5 in osteocytes.

### 2.5. Tumor-Suppressing Effect of TRAIL, p53, Chemerin, and Tumor-Promoting Effect of Nexin

We next examined the expression of the selected genes including Lrp5 (Wnt co-receptor), MMP2, MMP3, MMP9 (matrix metalloproteinases), Snail (inducer of epithelial-mesenchymal transition), Runx2, and TGFβ (bone-linked tumor promoters), as well as cleaved caspase 3 (apoptosis marker), in response to the selected regulatory proteins in EO771 mammary tumor cells. The administration of recombinant proteins of TRAIL, an apoptosis inducer, and the overexpression of p53, a tumor suppressor, reduced the tumorigenic genes and elevated cleaved caspase 3, an apoptosis marker (Figure 5A,B). The same response was observed to the administration of chemerin recombinant proteins, while the opposite response by RNA interference with chemerin siRNA (Figure 5C,D). Consistently, the responses to nexin recombinant protein and nexin siRNA were contrary to those to chemerin counterparts (Figure 5E,F). Regarding TRAMP prostate tumor cells, the trend in their responses to dopamine, cholesterol, TRAIL, p53 plasmids, chemerin, chemerin siRNA, nexin, and nexin siRNA was identical to that of EO771 cells (Appendix A).

### 2.6. Lrp5 as a Tumor-Suppressor in Osteocytes

To further evaluate the role of Lrp5 in tumor-osteocyte interactions in vitro, we employed Lrp5 plasmids and siRNA. In a spheroid competition assay, the co-culturing of EO771 tumor spheroids with osteocyte spheroids induced the shrinkage of tumor spheroids. Overexpression of Lrp5 in osteocytes enhanced the shrinkage of tumor spheroids (Figure 6A–D). Lrp5-overexpressing osteocyte-derived conditioned medium (CM) also inhibited EdU-based proliferation and scratch-based migration of MDA-MB-231 breast cancer cells and EO771 mammary tumor cells (Appendix A). Furthermore, Lrp5-overexpressing osteocyte-derived CM downregulated Lrp5, MMP9, Runx2, TGFβ, Snail, and nexin with an increase in chemerin in these cells (Figure 6E,F). Consistently, RNA interference with Lrp5 siRNA reversed the response of these genes (Figure 6G–I).

### 2.7. Regulation of Osteoclastogenesis by Chemerin and Nexin

In tumor-driven osteolysis, osteoclasts are responsible for bone resorption. We thus examined the effect of chemerin and nexin on the development of osteoclasts. In response to chemerin, the levels of NFATc1, a master transcription factor of osteoclastogenesis, and cathepsin K, a proteinase involved in bone resorption, were reduced in receptor activator of nuclear factor kappa-beta ligand (RANKL)-stimulated RAW264.7 pre-osteoclasts, while the response was opposite to the treatment with nexin (Figure 7A,B). Furthermore, the number of tartrate-resistant acid phosphate (TRAP)-positive multi-nucleated osteoclasts was reduced by chemerin and elevated by nexin (Figure 7C). Collectively, chemerin acted as a suppressor of tumor progression and osteoclastogenesis, whereas nexin as a promoter of tumor growth and osteoclast maturation.

## 3. Discussion

This study presented that loading-driven tumor suppression was mediated by both Lrp5-independent and Lrp5-dependent mechanisms. The Lrp5-independent mechanism was supported by the observation that regardless of Lrp5 presence in osteocytes, knee loading inhibited tumor growth in the mammary fat pad and bone loss in the tumor-invaded tibia. The Lrp5-dependent mechanism was evident since Lrp5-overexpressing osteocytes inhibited the growth of tumor spheroids and Lrp5-silenced osteocytes upregulated tumorigenic genes such as Runx2, MMP9, Snail, and TGFβ. Because these two mechanisms are present in the wild-type mice, the degree of tumor suppression in the tumor-invaded tibia of the wild-type mice was more significant than that of the knockout mice. In support of the Lrp5-independent mechanism, knee loading elevated dopamine and reduced cholesterol in the serum and urine of wild-type and knockout mice. Dopamine was reported to inhibit the oncogenic behavior of tumor cells, while cholesterol was reported to enhance it [23]. The signaling analysis showed that dopamine elevated the levels of tumor suppressors such as chemerin, p53, and TRAIL and reduced the level of nexin, a tumor promoter. By contrast, chemerin was downregulated and nexin was upregulated by cholesterol. Collectively, the present study demonstrated that the Lrp5-dependent and independent loading-driven anti-tumor pathways existed, and both pathways were linked to the upregulation of tumor-suppressing chemerin and the downregulation of tumor-promoting nexin (Figure 7D).

The dual pathway mechanism for loading-driven tumor suppression is different from the mechanism for loading-driven bone formation. Osteocytes are considered mechano-sensors in the bone, and it is reported that loading-driven bone formation is promoted by Lrp5-mediated Wnt signaling [24,25]. In this study, we also showed that the loading-driven increase in the bone volume ratio and bone mineral density was observed only in wild-type mice. Notably, however, the loading-driven suppression of tumor growth and tumor-induced osteolysis was observed in both wild-type and knockout mice. The result indicates that Lrp5 in osteocytes is necessary for loading-driven bone formation. For loading-driven tumor suppression, its contribution is significant through tumor-osteocyte interactions, but its absence does not eliminate the loading benefits because of the loading-driven systemic regulation of dopamine and cholesterol. Of note, the status of hormone receptors, such as estrogen receptors, can be involved in the responses to mechanical loading [26,27]. While Lrp5 in tumor cells acts as a tumor promoter in Wnt /β-catenin signaling that is considered a therapeutic target [28,29], the result in this study indicates that Lrp5 in osteocytes serves as a tumor suppressor. The overexpression of Lrp5 in osteocytes granted the tumor-suppressing capability to osteocytes. The co-culturing of tumor spheroids with Lrp5-overexpressing osteocyte spheroids shrank tumor spheroids, and their CM downregulated MMP9, Runx2, TGFβ, and Snail. It also inhibited the expression of nexin in tumor cells, while it elevated chemerin, a tumor-suppressor. Conversely, RNA interference with Lrp5 siRNA in osteocytes reversed the responses in tumor cells with a decrease in chemerin and an increase in nexin. Taken together, the role of Lrp5 in Wnt signaling differs in tumor cells and osteocytes, and thus the therapeutic inhibition of Lrp5/Wnt signaling may weaken the intrinsic tumor-suppressing capability of osteocytes.

The elevation of dopamine by physical activities is reported in mouse and human studies [30], but the mechanism of the loading-driven regulation of dopamine, as well as cholesterol, is not well understood in the Lrp5-independent pathway. Dopamine is a neurotransmitter [31], which is synthesized by dopamine neurons in the ventral tegmental area with a rate-limiting enzyme, tyrosine hydroxylase. While dopamine in this study was induced by mechanical loading, it is reported that electrostimulation can also elevate the level of dopamine [32]. The result of this study is consistent with the previous work in which the levels of dopamine and cholesterol were altered in human urine samples after step aerobics [23]. Of note, the physiological levels of dopamine and cholesterol in human serum are reported to be ~40 pg/mL and ~5 mM, respectively [33,34,35]. The levels we observed for mice in this study were on the same order but slightly higher than the human levels. In the Lrp5-independent mechanism, it is possible that besides dopamine and cholesterol, other neurotransmitters and metabolites are also involved as systemic tumor-regulating agents in the serum and urine.

The protein array analysis for cytokines and chemokines revealed that the levels of chemerin and nexin were mainly altered by knee loading in the serum of wild-type and knockout mice, as well as the responses to dopamine and cholesterol in tumor cells. It is reported that chemerin suppresses hepatocellular carcinoma metastasis, breast cancer growth, prostate cancer progression, skin carcinogenesis, and melanoma [36,37,38,39,40], while nexin is a stimulator of peritoneal metastasis in ovarian cancer [41], as well as the growth, migration, and invasion of breast cancer cells [42]. It is also reported that Wnt signaling in tumor cells is inhibited by chemerin while nexin is a downstream target of Wnt signaling [43,44,45]. We also observed that the levels of p53 and TRAIL were elevated by dopamine and reduced by cholesterol, in which p53 is a well-known tumor suppressor [46], and TRAIL is an apoptosis inducer of tumor cells [47]. Taken together, this study indicates that the upregulation of chemerin, TRAIL, and p53 as well as the downregulation of nexin by mechanical loading contributes to Lrp5-dependent and independent mechanisms by downregulating Lrp5, MMP2, MMP3, MMP9, Runx2, Snail, and TGFβ in tumor cells.

While this study employed two cancer models using breast and prostate cancer cells, the result may depend on individual types of cancer cells. It is also recommended to evaluate the efficacy of osteocyte-derived CM not only on breast and prostate cancers but also on other cancers. We also observed that the effects of chemerin and nexin on the development of bone-resorbing osteoclasts are consistent with their role in tumor progression. Chemerin suppressed osteoclastogenesis by inhibiting NFATc1 and cathepsin K, while nexin promoted the expression of these two genes. Collectively, the present study showed that chemerin suppresses both tumor progression and RANKL-induced osteoclastogenesis [48], while nexin exerted the opposite responses [49].

## 4. Materials and Methods

### 4.1. Cell Culture

EO771 murine mammary tumor cells (CH3 BioSystems, Amherst, NY, USA, RRID:CVCL_GR23) [50], TRAMP-C2ras murine prostate tumor cells (ATCC, Manassas, VA 20110, USA, RRID:CVCL_3615) [51], and MDA-MB-231 human breast cancer cells (ATCC, RRID:CVCL_0062) [52] were grown in DMEM. MLO-A5 murine osteocytes (obtained from Dr. L. Bonewald at Indiana University, Indiananpolis IN, USA, RRID:CVCL_0P24) and RAW264.7 murine pre-osteoclast cells (ATCC, RRID:CVCL_0493) were grown in αMEM. The culture media were supplemented with 10% fetal bovine serum and 1% antibiotics (1 × 10^2^ Units/mL penicillin, and 1 × 10^2^ ug/mL streptomycin; Life Technologies Corporation, Carlsbad, CA, USA), and cells were maintained at 37 °C and 5% CO_2._ Tumor cells were treated with dopamine (Tocris, Minneapolis, MN, USA), cholesterol (Sigma, Saint Louis, MO, USA), Chemerin recombinant protein, Nexin recombinant protein (R&D Systems, Minneapolis, MN, USA), and TRAIL recombinant protein (Biolegend, San Diego, CA, USA).

### 4.2. MTT, EdU, Invasion, Wound-Healing Scratch, and 3D Spheroid Competition Assay

Cell viability was examined using an MTT assay (Invitrogen, Carlsbad, CA, USA), and cell proliferation was evaluated by a 5-ethynyl-2’-deoxyuridine (EdU) assay using a fluorescence-based cell proliferation kit (Thermo-Fisher, Waltham, MA, USA) [53]. A Transwell invasion assay and a wound-healing scratch assay were conducted as described previously [54]. A three-dimensional spheroid competition assay was conducted by generating spheroids from EO771 and MLO-A5 cells in ultra-low attachment 96-well plates at concentrations of 1 × 10^4^ cells/well. After 24 h, the EO771 tumor spheroid was transferred to the well of the MLO-A5 osteocyte spheroid. Spheroids were fluorescently stained (Click-iT™ EdU Alexa Fluor™ 488 Imaging Kit; Thermo-Fisher, Waltham, MA 02451, USA).

### 4.3. Western Blotting, RNA Interference, Plasmid Transfection

Western blot analysis was conducted using the procedure previously described [55]. We used antibodies against Lrp5 (5731s, RRID:AB_10705602), MMP2 (87809s, RRID:AB_2800107), Runx2 (8486s, RRID:AB_10949892), Snail (3879s, RRID:AB_2255011), TGFβ (3711s, RRID:AB_2063354), cleaved caspase 3 (9661s, RRID:AB_2341188) (Cell Signaling, Danvers, MA, USA), MMP3 (sc-21732, RRID:AB_627958), MMP9 (sc-393859), cathepsin K (sc-48353, RRID:AB_2087687), NFATc1 (sc-7294, RRID:AB_2152503) (Santa Cruz, Dallas, TX, USA), Nexin (ab222754), Chemerin (ab103153, RRID:AB_10861013) (Abcam, Cambridge, MA, USA), p53 (UJ290170, Invitrogen, Carlsbad, CA, USA), TRAIL (NB500220, NOVUS, Centennial, CO, USA, RRID:AB_10003305), and β-actin (A5441, Sigma-Aldrich, Saint Louis, MO, USA, RRID:AB_476744). RNA interference was conducted using siRNA specific to chemerin (#82104), nexin (#150176), and Lrp5 (s69315) (Thermo-Fisher, Waltham, MA 02451, USA), with a negative siRNA (Silencer Select #1, Thermo-Fisher) using the procedure previously described [56]. The overexpression of Lrp5 and p53 was achieved by transfecting Lrp5 plasmids (#115907, Addgene, Cambridge, MA, USA, RRID:Addgene_115907), and p53 plasmids (#69003, Addgene, Cambridge, MA, USA, RRID:Addgene_69003), respectively.

### 4.4. ELISA and Protein Array Analysis

The levels of cholesterol and dopamine in the mouse urine and serum were determined using ELISA kits (MyBioSource, San Diego, CA, USA). We also employed a mouse XL cytokine array (R&D Systems, Minneapolis, MN, USA) and determined the expression levels of 111 cytokines and chemokines in the serum of the wild-type and knockout mice, as well as the protein extracts of EO771 mammary tumor cells.

### 4.5. Osteoclast Differentiation Assay

Using RAW264.7 pre-osteoclast cells (ATCC), an osteoclast differentiation assay was conducted in 12-well plates. During a seven-day assay, cells were grown with 20 ng/mL of RANKL, and the culture medium was exchanged once on day 4. Adherent cells were then fixed and stained with a tartrate-resistant acid phosphate (TRAP)-staining kit (Sigma-Aldrich, Louis, MO, USA), according to the manufacturer’s instructions. TRAP-positive multinucleated cells (above 3 nuclei) were identified as mature osteoclasts and counted.

### 4.6. Animal Models

The experimental procedures were approved by the Indiana University Animal Care and Use Committee (protocol code: SC292R; date of approval: 30 May 2019) and complied with the Guiding Principles in the Care and Use of Animals endorsed by the American Physiological Society. C57BL/six mice lacking Lrp5 in osteocytes (Dmp1-Cre; Lrp5f/f, RRID:MGI: 5014233) were created by breeding Dmp1-Cre transgenic mice with Lrp5 floxed mice, both of which have been described earlier [57]. Mice were housed four per cage and provided with mouse chow and water ad libitum.

In the mouse mammary tumor model, C57BL/six wild-type and knockout female mice (~six weeks, Envigo RMS, Inc., Indianapolis, IN, USA) received subcutaneous injections of EO771 cells (3 × 10^5^ cells in 50 μL PBS) to the mammary fat pad on day one. The animals were sacrificed on day 14, and the weight of each tumor was measured. In the wild-type and knockout mouse model of tibial osteolysis, C57BL/six male mice received an injection of TRAMP cells (3.0 × 10^5^ cells in 20 μL PBS) to the left tibia as an intra-tibial injection, and C57BL/six female mice received an injection of EO771 cells (2.5 × 10^5^ cells in 20 μL PBS). The animals were sacrificed on day 14. We harvested the tibiae for histology and μCT imaging, and the blood for ELISA.

### 4.7. Knee Loading

Knee loading was applied daily using an ElectroForce device (TA Instruments, New Castle, DE, USA). Mice were randomly assigned to the placebo and loading groups. Mice in the loading group were anesthetized with ~1.5% isoflurane, and sinusoidal loads of 1 N (peak-to-peak) at 2 Hz were given to the left knee for 5 min. Mice in the placebo group were anesthetized and placed on the loading device without receiving dynamic loads. No adverse effects in response to knee loading were observed.

### 4.8. μCT Imaging and Histology

The tibiae were harvested for μCT imaging and histology. Micro-CT was performed using Skyscan 1172 (Bruker-MicroCT, Kontich, Belgium). Using manufacturer-provided software, scans were performed at pixel size 8.99 μm, and the images were reconstructed (nRecon v1.6.9.18, Bruker-MicroCT) and analyzed (CTan v1.13, Bruker-MicroCT). We focused on the proximal tibia, 1 mm thick along the length of the tibia, distal to the growth plate. In histology, H&E staining and immunohistochemistry were conducted as described previously [54,58]. The samples were blinded for data analysis.

### 4.9. Statistical Analysis

For cell-based experiments, three or four independent experiments were conducted, and data were expressed as mean ± S.D. In animal experiments, the sample size in the mouse model was chosen to achieve a power of 80% with *p* < 0.05. The primary experimental outcome was tumor weight for the mammary fat pad experiment and the bone volume ratio (BV/TV) for the tibia experiment. Statistical significance was evaluated using a one-way analysis of variance. Post hoc statistical comparisons with control groups were performed using Bonferroni correction with statistical significance at *p* < 0.05. The single and double asterisks in the figures indicate *p* < 0.05 and *p* < 0.01, respectively.

## 5. Conclusions

Our work demonstrates the function of mechanical loading in the suppression of tumor growth; however, the study has a few limitations. We employed three tumor cell lines—two for breast cancer and one for prostate cancer—but the responses to mechanical loading may depend on the types of cancers and the status of hormone receptors of breast cancers. Since the bone size is significantly different between rodents and humans, the loading conditions such as loading forces need to be reevaluated for clinical translations. Further studies are necessary to evaluate the role of the nervous and endocrine systems in the regulation of dopamine and cholesterol in response to mechanical loading. In conclusion, we observed Lrp5-dependent and independent mechanisms for loading-driven tumor suppression. This study revealed that Lrp5 in osteocytes is beneficial to reduce tumor growth in the bone, but loading-driven suppression of tumor growth can take place even in the absence of Lrp5 in osteocytes via the regulation of dopamine and cholesterol. Five-min knee loading sharply altered the levels of cholesterol and dopamine in the mouse serum and urine. The regulation of dopaminergic and cholesterol signaling might provide a novel strategy to restrain tumor growth in breast cancer and bone metastasis associated with breast cancer and prostate cancer. The results indicate the possibility of developing a novel loading-driven and osteocyte-assisted approach for treating bone loss associated with breast and prostate cancers.

## Figures and Tables

**Figure 1 cancers-13-00267-f001:**
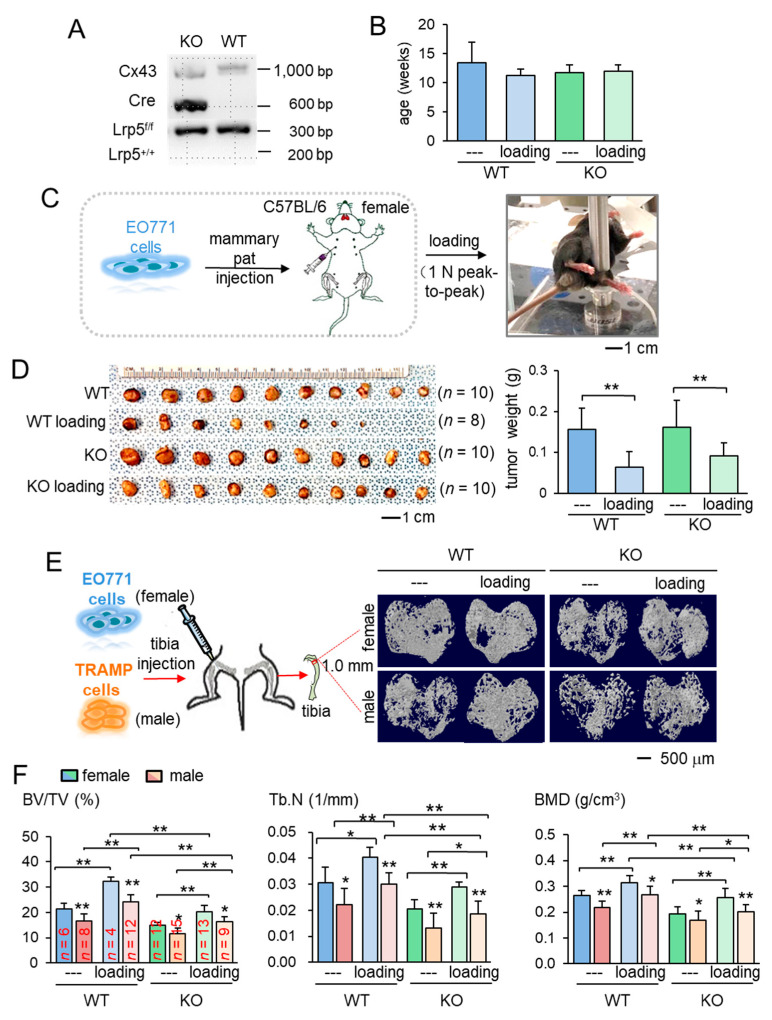
Loading-driven reduction in tumor progression. KO = knockout, and WT = wild-type. The single and double asterisks indicate *p* < 0.05 and *p* < 0.01, respectively. (**A**) Genotyping of wild-type and knockout mice. Cre was driven by the BMP1 promoter, and C x 43 was used as an internal control. The genotypes of a pair of mice (KO and WT) were shown based on amplified PCR products, including floxed Lrp5 (f/f), and Lrp5 (+/+) alleles. (**B**) Age of the mice used in this study. (**C**) Procedure for the inoculation of EO771 mammary tumor cells and cyclic knee loading. (**D**) Images of mammary tumors in four groups (wild-type and knockout with and without knee loading). (**E**) Micro CT images of the proximal tibiae of the wild-type and knockout mice with and without knee loading. Female mice received EO771 mammary tumor cells, while male mice received transgenic adenocarcinoma of the mouse prostate (TRAMP) prostate tumor cells in their proximal tibia. (**F**) Three micro CT-linked parameters in the proximal tibiae in the wild-type and knockout mice. BV/TV = bone volume ratio, Tb.N = trabecular number, and BMD = bone mineral density separation.

**Figure 2 cancers-13-00267-f002:**
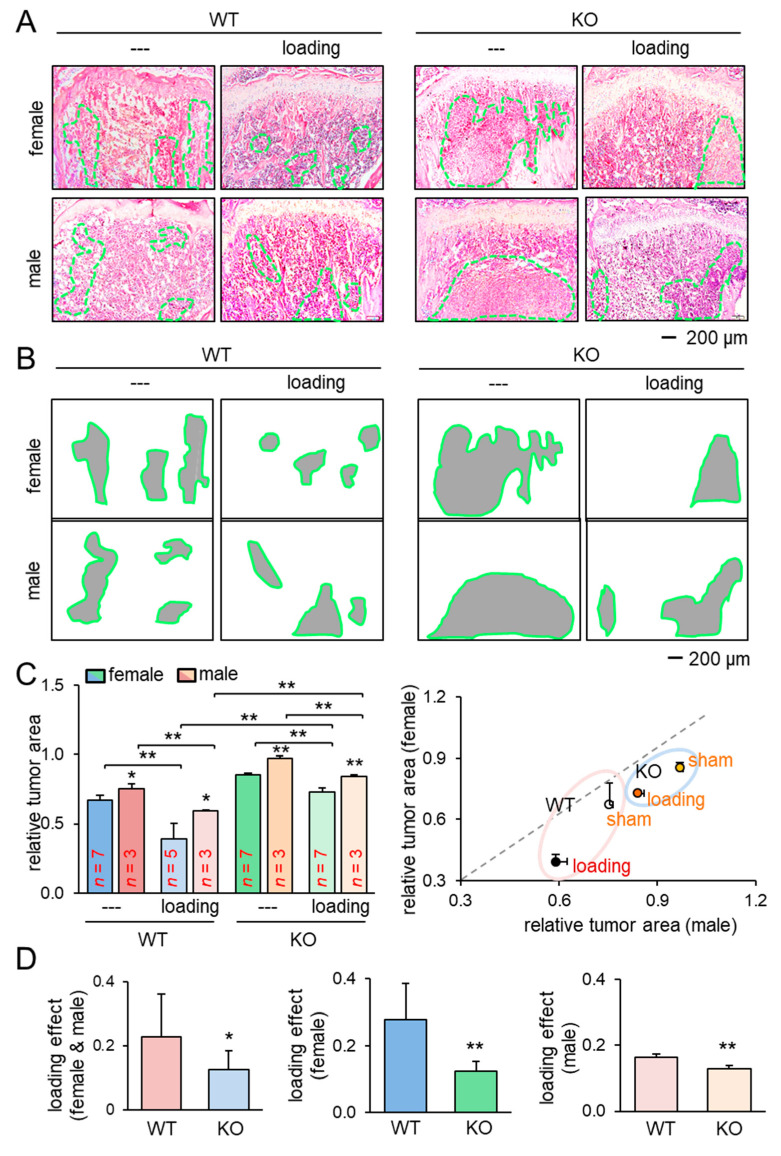
Reduction of the tumor-invaded area by knee loading. KO = knockout, and WT = wild-type. The single and double asterisks indicate *p* < 0.05 and *p* < 0.01, respectively. (**A**,**B**) H&E-stained histological images of the proximal tibiae of the wild-type and knockout mice with and without knee loading, and the region of the tumor-invaded area. (**C**) Comparison of the relative tumor area in the proximal tibiae for female and male mice. (**D**) Comparison of the loading effects on tumor areas between wild-type and knockout mice. The first plot is for all mice, and the second and third plots are for female and male mice, respectively.

**Figure 3 cancers-13-00267-f003:**
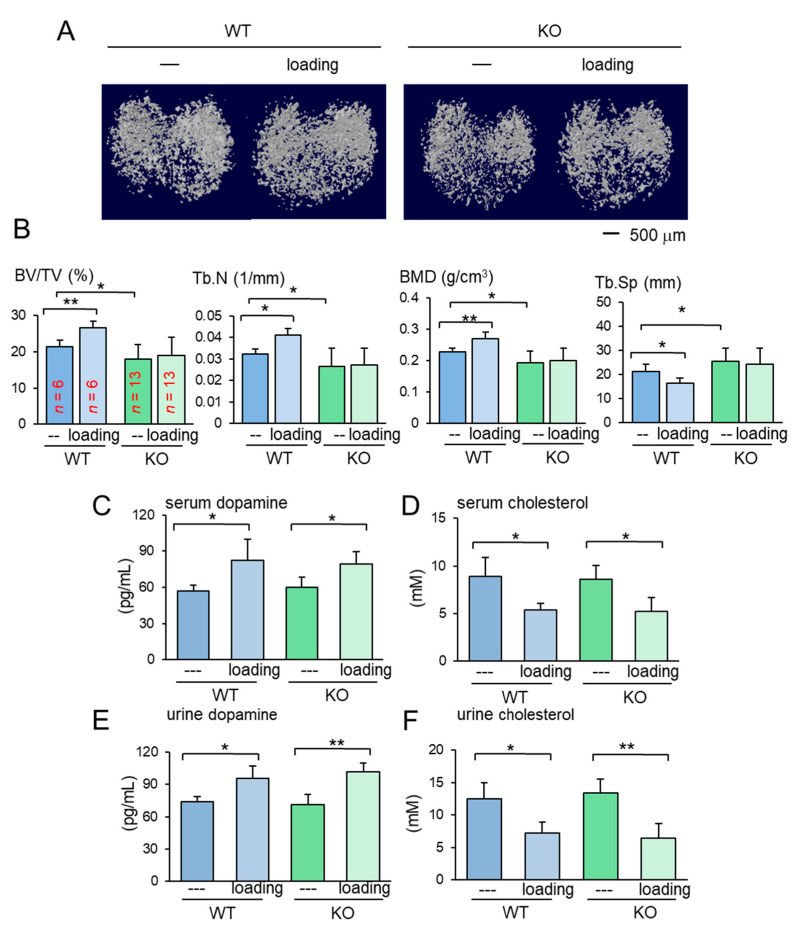
Loading effects on the proximal tibia for the wild-type and knockout mice. KO = knockout, and WT = wild-type. The data are expressed as the mean ± S.D. The single and double asterisks indicate *p* < 0.05 and 0.01, respectively. (**A**) Micro CT images of the proximal tibiae of the wild-type and knockout mice with and without knee loading. No tumor inoculation was applied. (**B**) Four micro CT-linked parameters in the proximal tibiae in the wild-type and knockout mice without tumor inoculation. BV/TV = bone volume ratio, Tb.N = trabecular number, BMD = bone mineral density, and Tb.Sp = trabecular separation. (**C**,**D**) Levels of dopamine and cholesterol in the serum in response to tumor inoculation. (**E**,**F**) Levels of dopamine and cholesterol in the urine in response to tumor inoculation.

**Figure 4 cancers-13-00267-f004:**
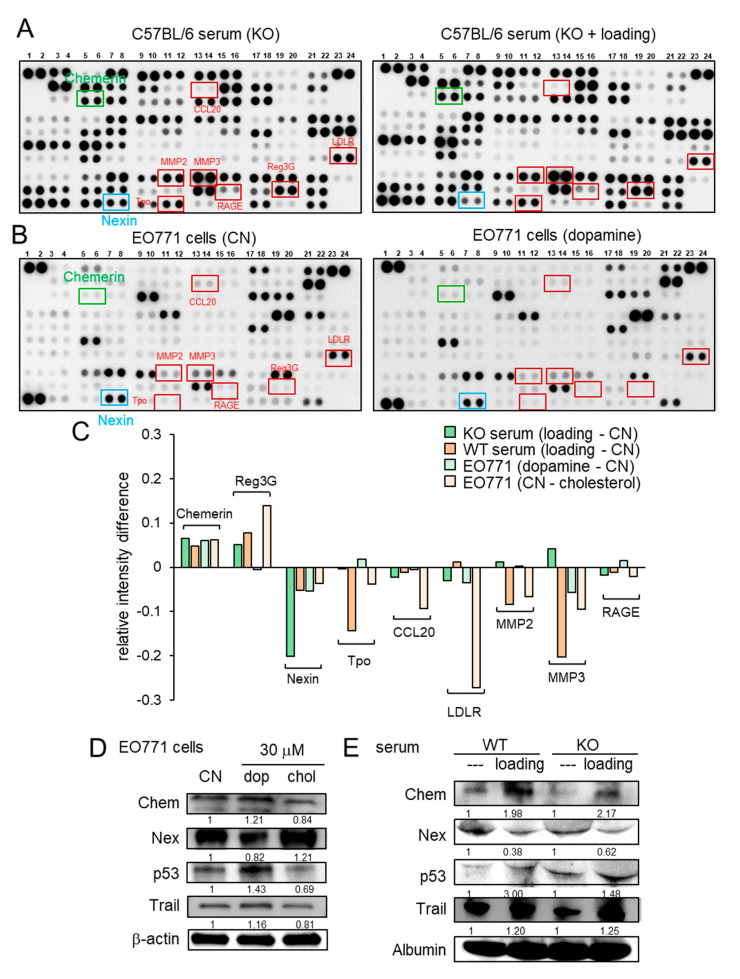
Cytokine array analysis using the serum and EO771 mammary tumor cells. WT = wild-type, KO = knockout, CN = control, dop = dopamine, chol = cholesterol, Chem = chemerin, and Nex = nexin. (**A**) Cytokine expression profile in the serum of knockout mice with and without knee loading. (**B**) Cytokine expression profile in EO771 cells with and without the administration of 10 μM dopamine. (**C**) Summary of the cytokine expression profiles for four pairs of samples, including the sera from the wild-type and knockout mice, and EO771 cells treated with dopamine and cholesterol. (**D**) Expression of chemerin, nexin, p53, and TRAIL in EO771 cells in response to 30 μM dopamine and cholesterol. (**E**) Expression of chemerin, nexin, p53, and TRAIL in the serum of the wild-type and knockout mice with and without knee loading.

**Figure 5 cancers-13-00267-f005:**
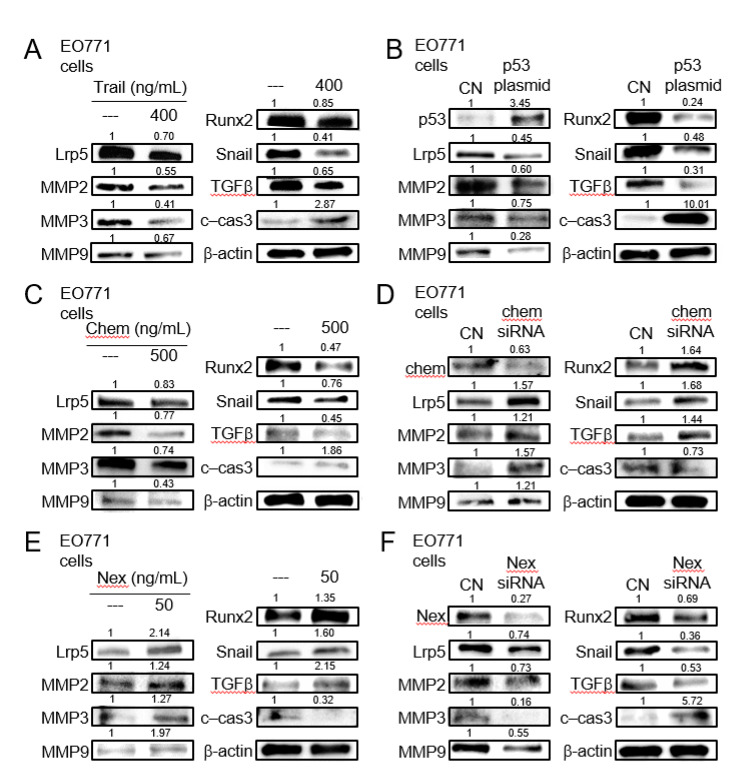
Expression of Lrp5, MMP2, MMP3, MMP9, Runx2, Snail, TGFβ, and cleaved caspase 3 in EO771 mammary tumor cells. CN = control, c-cas 3 = cleaved caspase 3, Chem = chemerin, and Nex = nexin. (**A**) Response to the administration of TNF-related apoptosis-inducing ligand (TRAIL). (**B**)Response to the transfection of p53. (**C**) Response to the administration of chemerin. (**D**) Response to RNA interference with chemerin siRNA. (**E**) Response to the administration of nexin. (**F**) Response to RNA interference with nexin siRNA.

**Figure 6 cancers-13-00267-f006:**
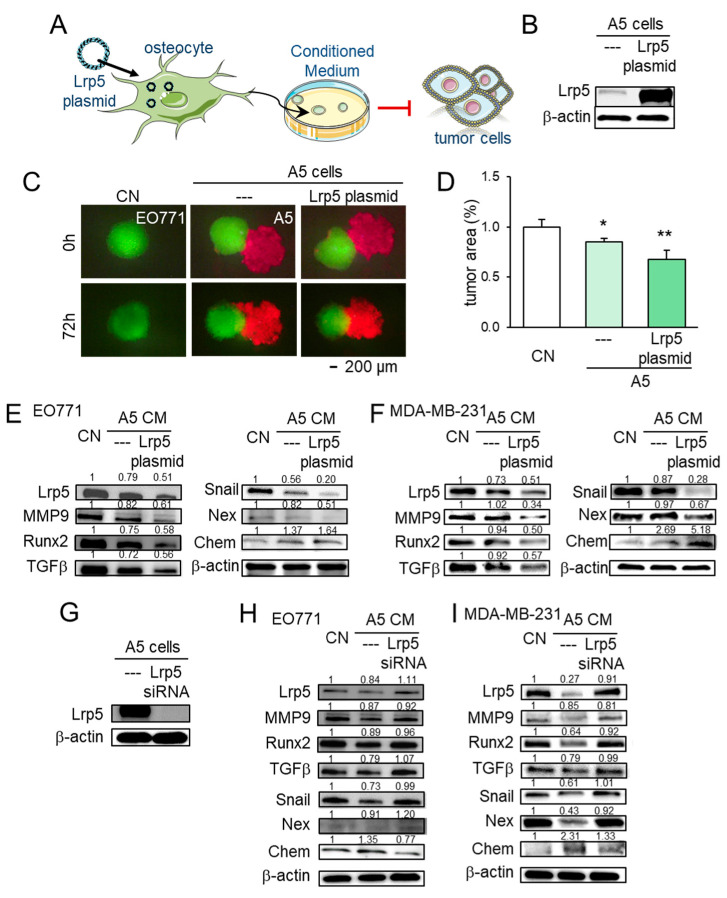
Effect of Lrp5 in osteocytes on EO771 and MDA-MB-231 breast cancer cells. The single and double asterisks indicate *p* < 0.05 and 0.01, respectively. CN = control, CM = conditioned medium, Chem = chemerin, Nex = nexin, cat K = cathepsin K. (**A**,**B**) Overexpression of Lrp5 in A5 osteocytes. (**C**,**D**) Shrinkage of EO771 tumor spheroids by the incubation with Lrp5-overexpressing osteocyte-derived CM and osteocytes. (**E**,**F**) Altered expression levels of Lrp5, MMP9, Runx2, TGFβ, Snail, nexin, and chemerin in EO771 and MDA-MB-231 cells in response to A5 osteocyte-derived CM with and without Lrp5 overexpression. (**G**–**I**) Altered expression levels of Lrp5, MMP9, Runx2, TGFβ, Snail, nexin, and chemerin in EO771 and MDA-MB-231 cells in response to A5 osteocyte-derived CM in the presence and absence of RNA silencing with Lrp5 siRNA.

**Figure 7 cancers-13-00267-f007:**
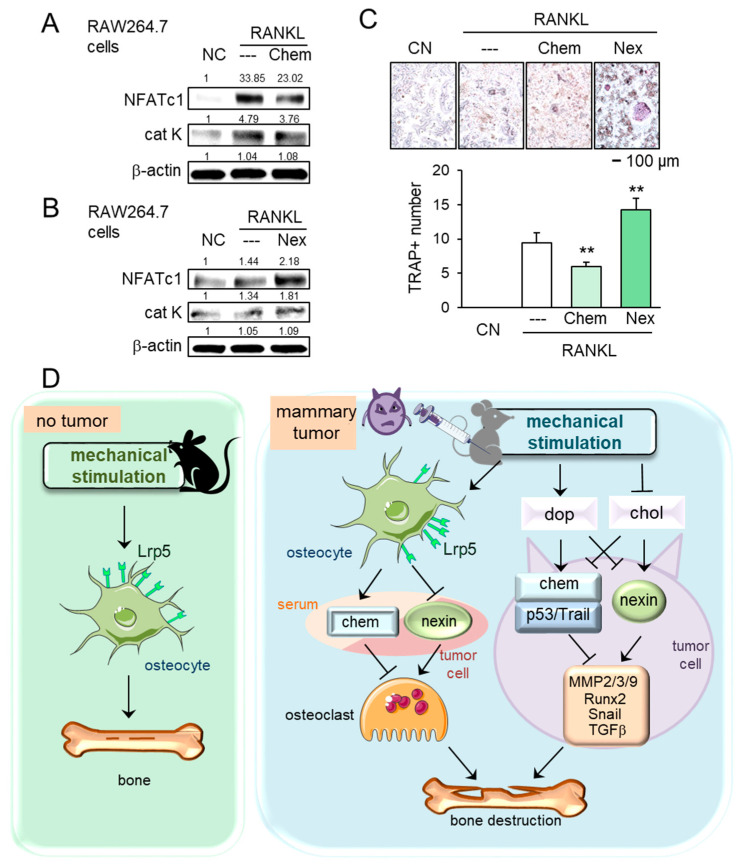
Effects of chemerin and nexin on osteoclasts, and the proposed regulatory mechanism. The double asterisk indicates *p* < 0.01. (**A**,**B**) Downregulation of NFATc1 and cathepsin K by chemerin and their upregulation by nexin in RANKL-stimulated RAW264.7 osteoclasts. (**C**) Reduction and elevation of tartrate-resistant acid phosphate (TRAP)-positive cells by chemerin and nexin, respectively. (**D**) Proposed regulatory mechanism for female mice in response to knee loading. Without tumor inoculation, Lrp5 in osteocytes is necessary to induce loading-driven bone formation. In the tumor-invaded tibia, mechanical loading suppresses bone destruction regardless of Lrp5 in osteocytes. The beneficial loading effects are mediated by the elevation of dopamine and the reduction in cholesterol. Chemerin and nexin act as a suppressor and promotor, respectively, to regulate oncogenic genes such as MMP2, MMP3, MMP9, Runx2, Snail, and TGFβ.

## Data Availability

The data presented in this study are available in this article (and Appendix A).

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
