# Peer review of "Mechanical Loading-Driven Tumor Suppression Is Mediated by Lrp5-Dependent and Independent Mechanisms"

_cancers, 2021, doi:10.3390/cancers13020267_

Round 1

Reviewer 1 Report

The manuscript concerns tumor suppression through mechanical knee loading in the presence and absence of Lrp5. From a series of experiments involving cancer cell lines and osteocyte-specific Lrp5 conditional knockout mice the authors conclude tumor suppression is caused by Lrp5 dependent and independent mechanisms.

Major Comments:

  1. Although suggested in the title and also in the abstract by the phrase “Lrp5-mediated Wnt signalling” the effects of Lrp5, either its overexpression or absence, on Wnt signalling are not extensively substantiated in the manuscript. Can the authors provide more convincing proof that Wnt signalling activity is affected?
  2. Figure 1D indicates there is a huge variation in the effects of mechanical loading on the size of mammary tumors with tumors being absent in some mice and clearly present in others. Do the latter mice still show lowered cholesterol and elevated dopamine levels? As the used mice are genetically homogeneous this is an unexpected result. The authors should address this variability and its possible origin(s) in the discussion section.
  3. Please give the appropriate gene names for chemerin and nexin at least once in the manuscript text.

Minor Comments:

  1. Page 1, line 27-28 – “..the question is whether Lrp5 tumor suppression.” Part of this sentence seems missing, please check and rephrase.
  2. Figure 1A – Please explain better in the legend and main text what exactly is shown is panel A, this is important as it deals with the KO mouse model used.
  3. Figure 2 – Please be clear how many mice were analyzed in this experiment and how many H&E stained sections were analyzed?
  4. Figure 3 – Be clear in the figure’s legend what the bar graphs indicate e.g. bar graph represent average values ±D. (n=xx).
  5. Legend Figure 4 – “..30 uM dopamine and cholesterol.” should be “..30 µM dopamine and cholesterol.”
  6. Page 12, section cell culture – Clearly indicate the whether the cell lines used are of murine or human origin.
  7. Page 13, line 303, 306 and 308 – Note that “5” should be in superscript.

Reviewer 2 Report

The authors examined the roles of LRP5 for the loading-driven bone formation, and tumor suppression using LRP5 KO mice. The authors conducted vary diverse experiment to reach critical conclusion, indicating that loading-driven tumor suppression and bone maintains were regulated by LRP5-dependent and independent mechanisms. The results are interesting. However, the reviewer found that the manuscript has weak points to be improved in the revised manuscript as following.

  1. The scope of the manuscript to big to fill the concept in a manuscript. Thus, the reviewer felt that data provided by the authors has not focused with mechanism study.
  2. Since the data is scattering to make fantastic story, the proof-based understanding of the manuscript was very difficult. Of course, the story jumping based on the previous literatures, which were found from different quarters, also can make readers delirious and could not recognize interesting values.
  3. There are some mis-translated results. For example, in Figure 1, although the biological significance between female and male apparently was not observable, the authors shows a asterisk with 0.05 p value. Since the error bars between female and male in graph F looked very big, the reviewer suggests to the authors that the data should be re-analyzed.
  4. In lane 112-114, the sentence is very confusing. The reviewer cannot understand how Fig 2D can conclude the bone protection or bone loss. If the authors want to say likely that, the results have to consider together with Fig 1E and 1F. Unfortunately, Fig. 1F BMD did not show the enough biological significance (p=0.08). This reviewer opinion is supported the statistical data as mentioned above.
  5. 3A and 3B, and S Fig 2 are confusing. In Fig. 3B, the error bars in sham of KO are too big to get biological significance compared to shams of WT. S Fig 2 is also same. Moreover, the bone uCT pictures of cortical regions as well as Fig 3A and S Fig 2 cannot say that these are objective. The pictures should be different from the depending on little difference place.
  6. The dopamine levels in Figure 3C and 3E are not clear whether the levels in the Figures are acceptable range or not. The same issues are applicable for the serum and urine cholesterol levels in Figure 3D and 3F. There are no any description in Introduction and Discussion section whether the levels of dopamine and cholesterol in the serum and urine. The reviewer suggests that the authors should added the information in discussion section for the issues.
  7. What does the Y-axis value? Log10 value? fold?, or %?. It is not clear the meaning of relative signal difference? How measure those? Where the data obtained from? in vitro or in vivo? Does the meaning of administration indicate in vivo?
  8. How serum sample can detect b-actin In Fig 4E? Thus, the results is not convincing. Moreover, It is not clear that all genes in Fig 5 are classified as tumorigenic genes or not. The authors described that the gene are tumorigenic genes in lane 147-148.
  9. LRP5-/- showed bigger tumor compared to wildtype in non-loading. If this case, trail should less compare to wildtype. Because of the basis of above rationale, loading showed less regression compared to wildtype. Unfortunately, trail levels were more increased LRP5 KO when loading was applied. Why? If trail levels were more increased, tumor size should become smaller.
  10. In Fig 5A, it makes me confuse. LRP5 is a upstream of Trail, or Trail is a upstream of LRP5. In Fig 4E, LRP5 KO shoed more increase of Trail protein levels, indicating that LRP5 is a upstream and negative regulator of Trail. However, In Fig 5, trail treatment suppressed LRP5 protein levels. If trail is a downstream of LRP5, the protein level of LRP5 could be not changed by trail treatment.

Reviewer 3 Report

The paper presented by Feng and distinguished colleagues reports interesting findings related to mechanical loading-driven tumor suppression and mediation by Lrp5/Wnt-dependent and independent mechanisms.  Although the major focus of the study (and text) is breast cancer, prostate cancer data are presented in some experiments as well.  In evaluating the Lrp5-mediated and Lrp5-non-mediated mechanisms of loading-driven tumor suppression, the authors investigated many different parameters including oncogenic genes and tumor-promoting cytokines and metalloproteinases.  The in vitro and in vivo models are state-of-the-art; however, the complexity of the study, together with the lack of appropriate rationale and discussion - - render the manuscript difficult to follow and the data presented difficult to interpret and appreciate.  The following comments are offered to strengthen the presentation and impact of the study:

  1. Abstract: Line 28, some words are missing relevant to the sentence finishing on this line.
  2. What is the rationale for the selection of the chosen oncogenic genes?
  3. What is the rationale for choosing the tumor-promoting cytokines and metalloproteinases?
  4. Why were TRAMP prostate tumor cells evaluated?
  5. How does one reconcile differences in hormone receptors in the models of breast and prostate cancer relevant to the focus of the study?
  6. Figure 6: (F) MDA-MB should be changed to the proper name/designation as MDA-MB-231.
  7. Figure 7: (D) This is a nice summary illustration, but it prompts the question of whether the mechanisms observed are exactly the same for breast and prostate cancer.
  8. The Discussion does not address similarities or differences between the tumor models.

In summary, this is an interesting and labor-intensive study that could benefit from more precise explanations and rationale for many of the parameters pursued.  The study is novel and is based on previous work from the authors in the field.  More emphasis should be placed on the translational relevance of the findings.

Round 2

Reviewer 1 Report

No further comments

Author Response

Thank you very much. We have checked the language and the spell in the manuscript.

Reviewer 2 Report

None

Author Response

(The authors gave the same response as above.)

Reviewer 3 Report

The authors have tried to address the concerns of the Reviewers -- and have made specific changes that partially clarified the questions raised. However, the major concern that persists relates to the inadequate reference to the prostate model (and prostate cancer) in: the Abstract, Keywords, Discussion and Summary. Why was the prostate model used in a study that seems to primarily focus on breast cancer?

Author Response

Response to Reviewer 3

The authors have tried to address the concerns of the Reviewers -- and have made specific changes that partially clarified the questions raised. However, the major concern that persists relates to the inadequate reference to the prostate model (and prostate cancer) in: the Abstract, Keywords, Discussion and Summary. Why was the prostate model used in a study that seems to primarily focus on breast cancer?

Thank you for the comment. As pointed out, the description in the previous revision was more focused on breast cancer, although we aimed to cover both breast cancer and prostate cancer. In our cell culture analyses in this study, we used not only EO771 mammary tumor cells and MDA-MB-231 breast cancer cells but also TRAMP-C2ras prostate cancer cells. We also employed female and male wild-type and conditional KO mice. In the analysis of tumor-invaded bone destruction, both female and male mice were employed and their responses to mechanical stimulation were evaluated. Despite these experimental results, we admit that the description of prostate cancer was not adequately provided in the previous revision.

In this revised manuscript, we added more descriptions of prostate cancer and the prostate model in the Simple Summary, Abstract, Introduction, Discussion, and Conclusions. We also included prostate cancer as a keyword and cited the following references in the main text.

  • A: Kamiya N, Suzuki H, Endo T, et al. Clinical usefulness of bone markers in prostate cancer with bone metastasis. Int J Urol. 2012;19: 968-979.
  • B: Castillejos-Molina RA, Gabilondo-Navarro FB. Prostate cancer. Salud Publica Mex. 2016;58: 279-284.
  • C: Rennier K, Shin WJ, Krug E, Virdi G, Pachynski RK. Chemerin Reactivates PTEN and Suppresses PD-L1 in Tumor Cells via Modulation of a Novel CMKLR1-mediated Signaling Cascade. Clin Cancer Res. 2020;26: 5019-5035.
  • D: Park SH, Eber MR, Shiozawa Y. Models of prostate cancer bone metastasis. Methods Mol Biol 2019; 1914: 295-308.
